# Inference is bliss: Simulation for power estimation for an observational study of a cholera outbreak intervention

**Ruwan Ratnayake**[1,2,3]*, **Francesco Checchi**[1], **Christopher I. Jarvis**[1,2], **W. John Edmunds**[1,2], **Flavio Finger**[3]

**1** Department of Infectious Disease Epidemiology, Faculty of Epidemiology and Population Health, London School of Hygiene and Tropical Medicine, London, United Kingdom, **2** Centre for the Mathematical Modelling of Infectious Diseases, London School of Hygiene and Tropical Medicine, London, United Kingdom, **3** Epicentre, Paris, France

* ruwan.ratnayake@lshtm.ac.uk

## Abstract

### Background

The evaluation of ring vaccination and other outbreak-containment interventions during severe and rapidly-evolving epidemics presents a challenge for the choice of a feasible study design, and subsequently, for the estimation of statistical power. To support a future evaluation of a *case-area targeted intervention* against cholera, we have proposed a prospective observational study design to estimate the association between the strength of implementation of this intervention across several small outbreaks (occurring within geographically delineated clusters around primary and secondary cases named 'rings') and its effectiveness (defined as a reduction in cholera incidence). We describe here a strategy combining mathematical modelling and simulation to estimate power for a prospective observational study.

### Methodology and principal findings

The strategy combines stochastic modelling of transmission and the direct and indirect effects of the intervention in a set of rings, with a simulation of the study analysis on the model results. We found that targeting 80 to 100 rings was required to achieve power ≥80%, using a basic reproduction number of 2.0 and a dispersion coefficient of 1.0–1.5.

### Conclusions

This power estimation strategy is feasible to implement for observational study designs which aim to evaluate outbreak containment for other pathogens in geographically or socially defined rings.

**Data Availability Statement:** No data was used for this study. The R code for the simulations is available here: https://github.com/ruwanepi/CATI-power-sim-shared.git.

**Funding:** RR is funded by a Doctoral Foreign Study Award from the Canadian Institutes of Health Research (award number DFS-164266, https://cihr-irsc.gc.ca/). FC CIJ, and WJE are funded by UK Research and Innovation as part of the Global Challenges Research Fund (grant number ES/P010873/1, https://www.ukri.org/). The funders had no role in study design, data collection and analysis, decision to publish, or preparation of the manuscript.

**Competing interests:** The authors have declared that no competing interests exist.

## Author summary

From Ebola virus disease outbreaks to the COVID-19 pandemic, the use of real-time evaluations of interventions to contain outbreaks is vital for rapidly estimating impact during the outbreak itself. Such evaluations must be both epidemiologically rigorous and logistically feasible to justify their conduct during an outbreak. In this short report, we report on the process (with R code) and the results of a simulation strategy that we devised for power estimation for a prospective observational study of a novel intervention ("case-area targeted intervention") to contain cholera case clusters that present at the start of a new outbreak. We used simulation in two ways: mathematical modelling to simulate the impacts of a cholera outbreak and the intervention, and simulation of the study analysis on the model results. The strategy provided estimates of the sample sizes of study units required to achieve 80% and 90% power. Our findings reinforce that this process is feasible to implement for similar observational study designs which aim to evaluate outbreak containment for other pathogens in geographically or socially defined rings.

## Introduction

Fast and efficient disease control approaches are critical for controlling cholera epidemics. Case area-targeted interventions (CATI) aim to interrupt transmission within small cholera outbreaks by rapidly addressing different routes of infection with multiple interventions (i.e., antibiotic chemoprophylaxis, household water treatment, and oral vaccination) in geographical 'rings' of 100–250 metres around the household of the index case. [1,2] Such containment strategies for small outbreaks target people at the highest risk of infection and may be less resource-intensive and more effective than mass, community-wide campaigns over large geographical areas. [1]

We designed an observational study to measure the effects of CATI during a future cholera epidemic response, to be conducted by Médecins Sans Frontières. The evaluation of CATI presents several challenges for the choice of a feasible study design and subsequently, for the estimation of statistical power. Randomizing individuals or communities to different interventions or a placebo is often not feasible and ethically problematic during a demanding epidemic response in a low-resource setting. For the evaluation of ring vaccination with a new vaccine during the 2016 Ebola epidemic in Guinea, an adapted cluster randomized-controlled trial (RCT) design was developed wherein each ring of contacts of confirmed cases was randomized to a different delay to implementation, thereby producing intervention and control groups. [3] During a cholera outbreak, where a package of routine rather than novel interventions is applied, the objective is to assess the allocation strategy. For this question, an RCT design may not always be appropriate or feasible.

Here, a prospective observational study design is considered, where participants or groups are not randomized and the outcome is measured prospectively. [4] In our example, the measurement of effectiveness (i.e., incidence) is related to the strength of implementation of the intervention across small outbreaks rather than an assigned presence or absence of the exposure (i.e., CATI). The strength of implementation is represented by the natural delay between case notification and the implementation of CATI, which may differ across several small outbreaks. This results in CATI rings categorized by the delay between case notification and implementation. Two interrelated challenges emerge, which do not fit well with a classical statistical approach for study design. First, the analysis does not conform to the conventional formulae for sample size and power estimation given the presence of several 'natural' control

groups. Second, the non-independence of infection risk between persons drives the incidence and is difficult to estimate *a priori*. [5] The interventions produce direct and indirect effects on infection and transmission, with infection prevention, infection, and/or treatment of one person affecting the outcome of another person. [6] Moreover, the cumulative effects of a package of interventions are difficult to predict.

In this report, we describe a strategy to estimate power for a prospective observational study across a range of sample sizes. This approach combines stochastic simulation modelling of small outbreaks and the direct and indirect effects of the intervention, with a simulation study of the study analysis based on model results. While simulation studies are often conducted to estimate power for RCTs, there is little documentation of (a) simulation used for other study designs and (b) mathematical modelling to simulate transmission dynamics for power estimation. [7–9] We provide details of the approach, and R code, as a foundation for further application to outbreak intervention studies of other pathogens.

## Methods

We describe the study design for which we are calculating power. We then describe the simulation study using the *Aim*, *Data Generating Mechanism*, *Estimand*, *Methods*, *Performance Measures* (ADEMP) framework for the coherent reporting of simulation studies. [7]

### Summary of the prospective observational study design

The impact of CATI (which includes single-dose oral cholera vaccination (OCV), point-of-use water treatment, and antibiotic chemoprophylaxis) will be measured by the reduction in the incidence of cholera around the index cases of small outbreaks through direct and indirect protection, as a function of the time to implementation of CATI (Fig 1). The intervention is triggered when a suspected case is detected and tests positive by an enriched rapid diagnostic test (RDT). [10] Then, a 100–250 metre radius around the index case's household is outlined

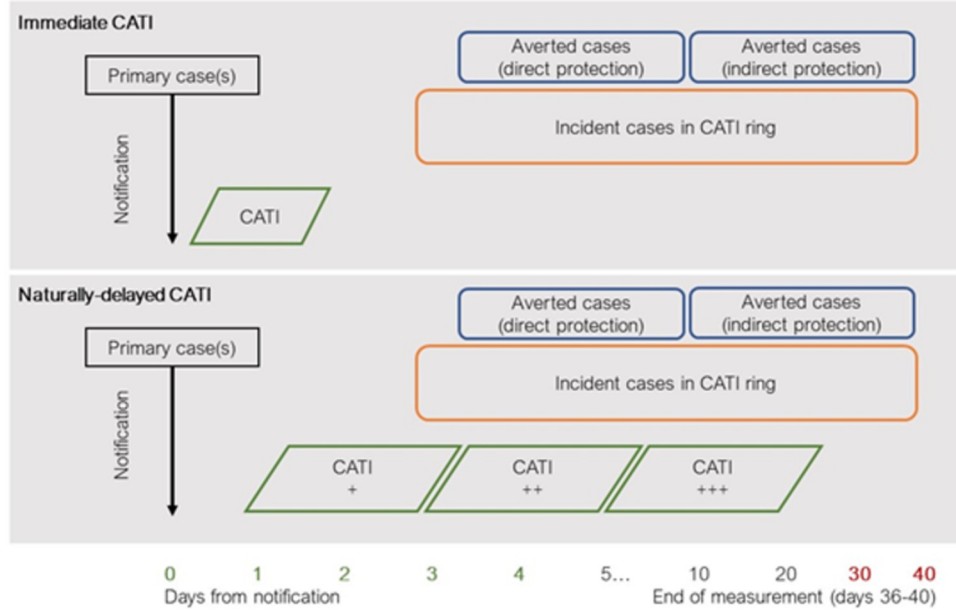

**Fig 1. Diagram of the study design.** Delays to implementation of CATI give rise to natural controls. A regression analysis is used to model the observed incidence of enriched rapid diagnostic test-positive cholera in rings (outcome) as a function of the delay to response. CATI = case-area targeted intervention.

(hereafter, the 'ring'), wherein CATI is rapidly implemented. While the first outbreak clusters may be responded to very rapidly, as the size of the epidemic increases logistical barriers for field teams are anticipated to result in delays to implementation in new rings of up to 7 days, thus creating natural control groups. The ring is the unit of analysis. A regression analysis will model the observed incidence of enriched RDT-positive cholera in rings relative to the time to response (in days) and coverage. The regression function quantifying the association between timeliness/coverage and incidence provides a measure of effectiveness at different levels of performance.

## Summary and rationale for the simulation

The study design rests on the assumption that the regression analysis will have sufficient power to detect an association between CATI performance and the incidence of cholera, i.e. that data from a sufficient number of rings of given size and characteristics (e.g. transmissibility of cholera within the rings) will be available. The **aim** of the simulation study is to explore these sample size requirements. We chose as a **data-generating mechanism** a stochastic transmission model to predict the incidence and the direct and indirect effects of CATI, applied with varied delays, on transmission across a large set of rings during a 30-day period. This mechanistic model of transmission and the predicted effect of the intervention is driven by transmission dynamics and is therefore more realistic than the assignment of an effect size, as typically used in statistical simulations. [5] The basic reproduction number for cholera, $R_0$, is varied in the modelling scenarios. The **estimand** is the incidence of suspected cholera in each ring in the first 30 days after presentation of the index case. The **method** used for simulation involves a regression applied to each set of simulated data to estimate the association of CATI performance and cholera incidence, while tracking *p-values* for the association. For each combination of simulation parameters, the mechanistic model and regression on the resulting simulated data are replicated by an assigned number of simulations ($n_{sim}$) by randomly sampling without replacement over the anticipated number of rings in the study ($n_{rings}$). The proportion of runs in which the regression yields a significant association provides a measure of power given that a sample of $n_{rings}$ are available. A range of $n_{sim}$ (1000–3000) is used to assess the stability of power estimates. The **performance measure** is the predicted power.

All analyses were carried out in R version 4.0.5, using the following packages: bpmodels [11] for branching process modelling, lme4 for generalized linear mixed modelling, and the map_dfr() function of the purrr package to repetitively apply functions for the simulation coding.[12] In the following sections, we describe each step in detail, which together with the code provided, can be used to replicate the simulation (https://github.com/ruwanepi/CATI-power-sim-shared.git).

## Stochastic transmission model

Using the bpmodels package, we applied a branching process model which generated infected persons and accounted for the depletion of susceptible persons to produce the incidence for each of 100,000 rings in the first 30 days after notification of the index case. The population size of the ring (normal distribution with mean 500, SD 50) was within the range of the number of people living in a 100–250 metre radius in major African cities including N'Djamena, Conakry, and Lumumbashi (mean 295, range 55–456 persons). [2] People were assumed to mix homogeneously. Given the efficiency of person-to-person and environmentally-mediated cholera transmission within households, there is potential for exponential growth, mediated by the depletion of susceptibles, before effective control measures are implemented. [13] We assumed that the first notified index case was the true primary case for the outbreak and that

all infectious cases were symptomatic and detectable, with some delay. Infection-to-reporting delays before and after CATI implementation were set as Poisson-distributed. The main outcomes of the model were, by ring, (a) cumulative incidence at day 30, and (b) a random effect accounting for the varying delay from infection to reporting of the primary case in the model (categorized as 0, 1, $\geq$1 days), as a proxy of the surveillance capacity by geographic area.

To model transmission, the parameters listed in Table 1 were used, and were either sampled from the underlying distributions or fixed. All persons were assumed to be susceptible without immunity derived from previous vaccination or exposure to *V. cholerae*. An outbreak started with a single seed case and each case generated a number of secondary cases drawn from a negative binomial distribution $Z \sim NegB\,(R_0, k)$. The mean is equal to the basic reproduction number at the early phase of the outbreak among an unvaccinated population ($R_0 = 1.5$, $R_0 = 2.0$). [14,15] Heterogeneity in the number of new infections produced by each individual is represented by a dispersion coefficient (D = 1.0, D = 1.5), which relates to the dispersion parameter of the negative binomial distribution, $k\left(k = \frac{R_0}{D-1}\right)$. [16,17] Each potential new infection was assigned a time of infection drawn from the serial interval distribution, *S ~ gamma (shape = 0.5, rate = 0.1)*. [14] The number of susceptible persons in the population was progressively reduced due to infection or immunity, reducing the mean of the negative binomial offspring distribution by a factor n/N, where n is the number of remaining susceptible and N is the total population, while keeping the dispersion coefficient constant, and truncating the distribution at n. We assumed that no other interventions were implemented before CATI. Four scenarios using high and low $R_0$ and $D$ were modeled (Table 1).

CATI interventions were then simulated, with a delay from notification of the index case as determined by a Poisson distribution, and the upper limit approximately based on the 75th percentile of the median delay from symptom onset to case presentation derived from a meta-

**Table 1. Parameters for the stochastic transmission model.**

| Parameter | Values | Reference |
|---|---|---|
| **Sampled** | **Mean (SD)** | |
| Serial interval, days | 5 (8), by negative binomial distribution | Azman et al, 2016[14] |
| Reporting delay (before CATI), days | 1 (0.9), by Poisson distribution ($\lambda$ = 1) | Assumed |
| Reporting delay (after CATI), days | 0.5 (0.7), by Poisson distribution ($\lambda$ = 0.5) | Assumed |
| Implementation delay, days | 3 (1.9), by Poisson distribution ($\lambda$ = 1.4) | Ratnayake et al, 2020 [18] |
| Population size of ring ± SD | 500 (50), by normal distribution | Finger et al, 2019[2] |
| **Fixed** | **Values** | |
| Basic reproduction number for index cases, $R_0$ | 1.5, 2.0 | Azman et al, 2016[14] Camacho et al, 2018[15] |
| Dispersion coefficient, *D* | 1.0, 1.5 | Emch et al, 2008[16] |
| Initial immune, persons, % | 0% | Assumed |
| Implementation duration, days | 1 (main analysis), 2 | Ouamba et al, 2021[20] |
| Population coverage, % | 80% (main analysis), 50%, 60%, 70% | Parker et al, 2017[19] |
| Efficacy of antibiotics, % | 66% | Reveiz et al, 2001[23] |
| Efficacy of water treatment, % | 26% | Fewtrell et al, 2005[21] |
| Efficacy of safe water storage, % | 21% | Roberts et al, 2001[22] |
| Efficacy of vaccination, % | 87% | Azman et al, 2016[24] |

During each simulation, sampled values are probabilistically sampled and fixed values remain constant. Median (SD), single values, or proportion efficacy are given. Efficacy measures are summarized in Ratnayake et al, 2021.[1]

analysis of cholera outbreaks (0 to 5 days), assuming that the surveillance set-up for CATI will prevent longer delays. [18] We assumed that implementation took one day and the population-based coverage was 80%. [19,20] CATI included distribution of (1) water, sanitation, and hygiene (WASH) materials including chlorine tablets and a narrow-neck container so that the efficacy in reducing bacterial concentration via household water treatment (26%) and safe storage (21%) remained consistent for the 30-day period (cumulative efficacy, 41.5%). [21,22]; single-dose, oral antibiotic chemoprophylaxis against infection so that the efficacy in preventing infection (66%) was maintained for the first 2 days, whereafter it loses effect due to its biological half-life [2,23]; and single-dose, oral cholera vaccination (OCV) prevented infection with an efficacy of 87% over a 2-month period, taking effect 7–11 days after administration when peak vibriocidal response is reached.[24] The effectiveness *(efficacy*coverage)* was calculated in three phases over the 30 days reflecting the plausible timespan over which the relative effects of each intervention would manifest: (1) days 1 to 2: WASH and antibiotic chemoprophylaxis, (2) days 3 to 6: WASH only, (3) days 7 to 30: WASH and vaccination. The combined effect of concurrent interventions was computed as *(1 - ((1-effect.A)* (1-effect.B)* . . .*(1-effect. Z)))*.

We conducted two sensitivity analyses to explore the main assumptions of rapid 1-day duration of implementation and population coverage of 80%. We evaluated a longer duration of implementation of 2 days and lower population-based coverage estimates of 50%, 60%, and 75%, based on findings from field studies. [19,20].

## Simulation method

A generalized linear mixed model (GLMM) was used to estimate the response variable (cumulative incidence rate) as a function of time to implementation. A negative binomial distribution accounted for overdispersion. Fixed effects quantified the main explanatory variable: the overall effect of time from case presentation to CATI implementation. The logarithm of the ring population was used as an offset to produce an incidence rate ratio (IRR). A random effect accounted for the delay from infection to presentation of the index case, which was categorized into 3 classes (0, 1, $\geq$1 days). For simplicity, other potential confounders that would require explicit measurement of geographical locations of rings were not considered (e.g., distance between the ring and the base of the field team). Model fit was assessed using the ratio of sum of squares of Pearson residuals to the residual degrees of freedom (to estimate overdispersion), inspection of the width of confidence intervals, and plotting of response by random effect levels (to estimate the benefit of including the random effect, as compared to using a generalized linear model (GLM)).

As the health of individuals in the same ring may be correlated, regression modeling approaches that account for the clustered nature of the data should be used for the study analysis. This includes GLMM, generalized estimating equations (GEE) and generalized additive models (GAM). GLMM uses random effects to account for contextual factors from the rings that alter the relationship between the exposure and the population effect, while GEEs infer the population-averaged effect across all rings. [25] As we expect there will be variance between rings and we may want to explore it further, GLMMs are preferred over GEEs for this study. GAMs add together the non-parametric and parametric fits of separate regressors into a transformed regression. [26] In this study, GAMs may be used if the observed relationship between delay to response and incidence offers a better fit than a purely parametric GLMM model. Regardless of model choice, the effect estimates should remain similar and unbiased.

The expected power was estimated for a range of sample sizes ($n_{rings}$ = 50–150 rings), based on recent CATI experiences during large epidemics in Nepal and Haiti, with a target of 80%

**Table 2. Parameters for the simulation study.**

| Parameter | Value | Reference |
|---|---|---|
| Number of rings produced by stochastic model | 100,000 | Assumed |
| Number of rings randomly sampled ($n_{rings}$) | 50, 75, 100, 125, 150 | Roskosky et al, 2019[27] Michel et al, 2019[28] |
| Number of simulations run for each value of $n_{obs}$ ($n_{sim}$) | 500, 1,000, 3,000 | Morris et al, 2019[7] |

power. [27,28] We simulated 100,000 CATI rings using the above-described method. We then conducted a simulation study by randomly sampling a set number of rings ($n_{rings}$), a set number of times ($n_{sim}$). A negative binomial GLMM was run on each set of $n_{rings}$. Power was assessed as the number of simulations with a significative effect of delay to CATI implementation *(p<0.05)*, considered to be true positives, divided by the number of $n_{sim}$. $n_{rings}$ was varied to assess the effect of the number of rings in each study on power. Table 2 lists the simulation parameters including the range of $n_{sim}$ values used to demonstrate consistency in results. For each set of $n_{rings}$ rings randomly sampled without replacement from the 100,000 rings simulated by the stochastic model, 500–3,000 simulations were run to evaluate the reliability of the results.

## Findings

Using $R_0$ = 2.0, D = 1.5 and $n_{sim}$ = 100,000, the mean caseload increased with each single day from 12 cases (with delays of 0 days) to 59 cases (with delays of 7 days). A higher proportion of outbreaks were extinct by day 30 for the ≤3-day category versus the >3-day category. An IRR of 1.27 (95% CI 1.25–1.29) was produced, demonstrating a 27% increase in the incidence rate per single day increase in the delay of implementation of CATI (visualized in Fig 2). The model fit is described in S1 Text.

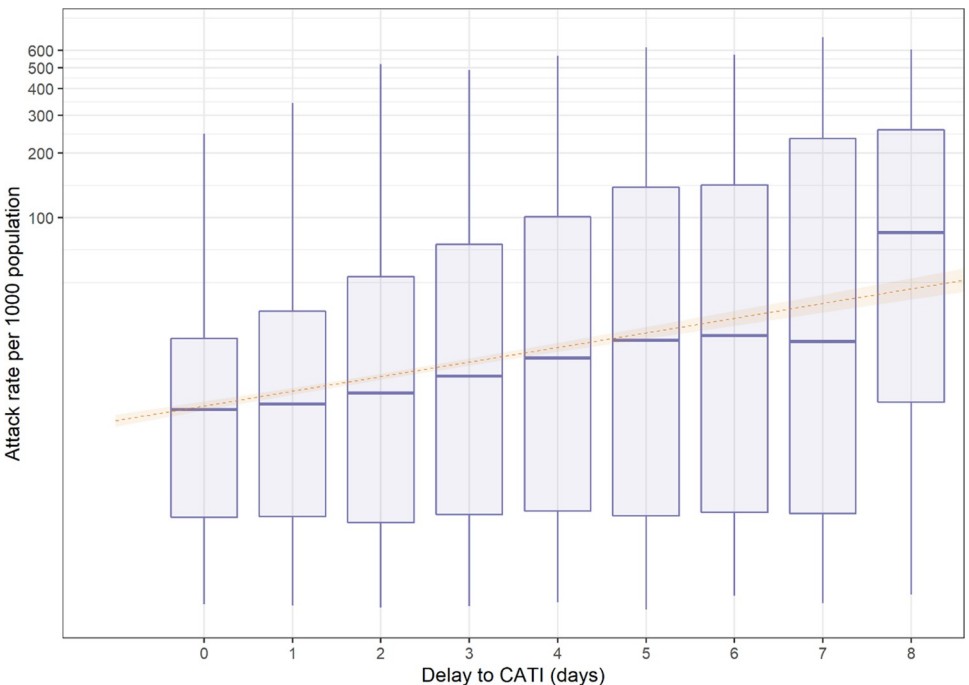

**Fig 2. Boxplots of the attack rate of cholera cases (per 1000 population, on a log10 scale) categorized by the delay to CATI implementation (in days) using 100,000 rings (with generalized linear model of the association outlined in orange).**

**Table 3. Power estimates from main simulations and sensitivity analyses.** Shading indicates the variable that was changed (grey), and where power estimates were farthest from the 80% target (≤69%, in orange), close to the target (≥70 to 79%, in light green), and at or above the target (≥80%, in dark green).

| $R_0$ | D | Duration | Coverage | Number of rings | | | | | |
|---|---|---|---|---|---|---|---|---|---|
| | | | | 50 | 75 | 80 | 100 | 125 | 150 |
| 2 | 1.5 | 1 | 80% | 52.4 | 71.7 | 73.7 | 80.6 | 88.7 | 94.7 |
| 2 | 1 | 1 | 80% | 57.3 | 77.1 | 81.2 | 85.8 | 92.7 | 96.2 |
| 1.5 | 1.5 | 1 | 80% | 33.6 | 37.4 | 49.5 | 49.7 | 58.3 | 62.8 |
| 2 | 1.5 | 2 | 80% | 44.4 | 60.4 | 60.5 | 69.5 | 78.7 | 84.7 |
| 2 | 1.5 | 1 | 50% | 52.9 | 64.2 | 68.9 | 77.5 | 85.9 | 92.4 |
| 2 | 1.5 | 1 | 60% | 51.3 | 64.6 | 70.6 | 76.8 | 85.3 | 92.3 |
| 2 | 1.5 | 1 | 75% | 53.6 | 68.0 | 72.8 | 79.1 | 84.7 | 92.1 |

The compiled power estimates are presented in Table 3 and the main power estimations where $R_0$, D, and $n_{rings}$ were varied are displayed in Fig 3 (additional graphs are found in Figs A—D in S1 Text). Using the main model ($R_0 = 2.0$ and D = 1.5), the simulation returned 80.6% (95% CI 71.2–87.6) power with $n_{rings} = 100$; 73.7% power with $n_{rings} = 80$; and, 88.7% power with $n_{rings} = 125$ (Fig 3). Using $R_0 = 2.0$ and D = 1.0, the simulation reached 81.2% (95%

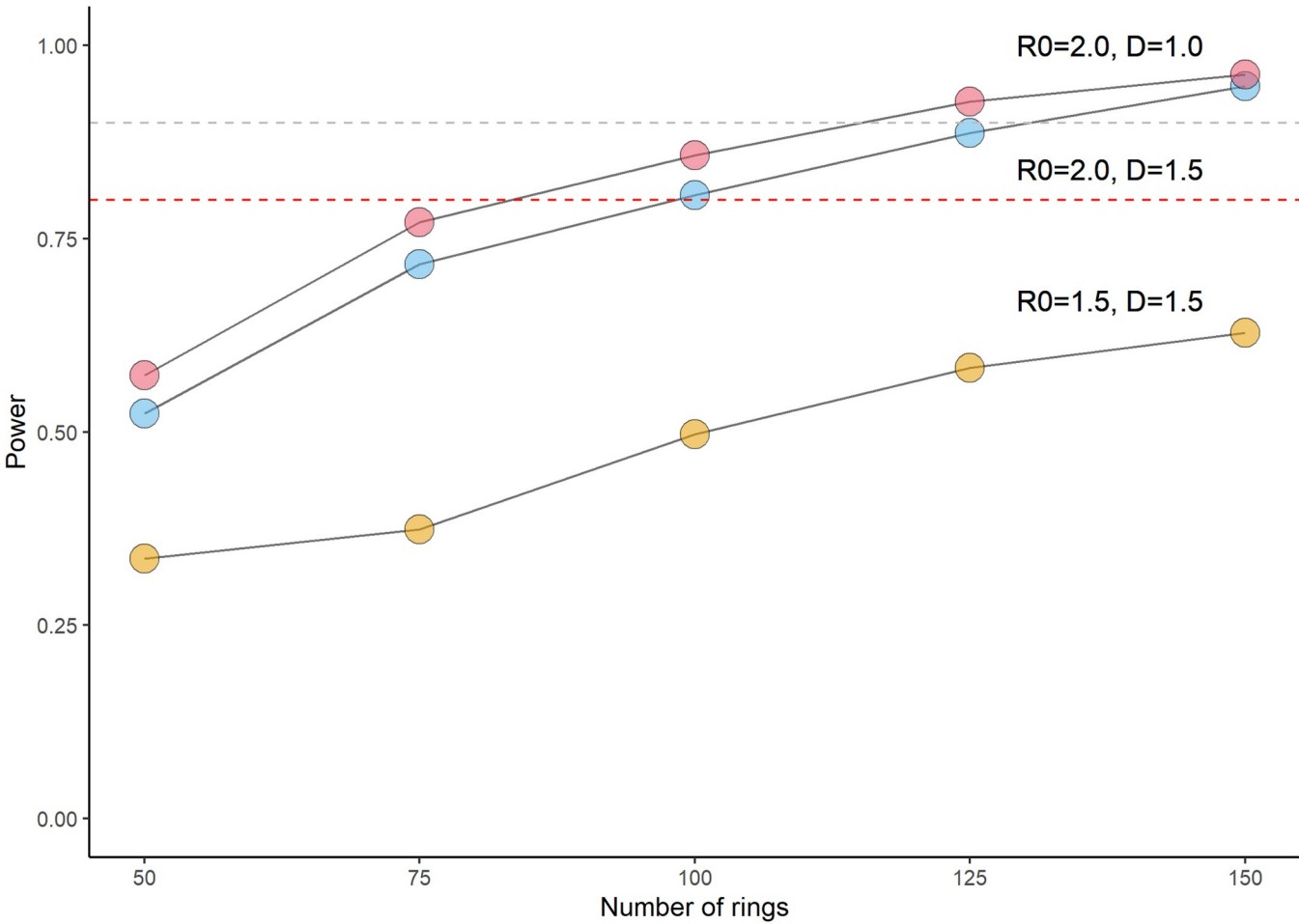

**Fig 3. Power estimation by the number of rings.** (A) $R_0 = 2.0$, D = 1.0 (RED), (B) $R_0 = 2.0$, D = 1.5 (BLUE), (C) $R_0 = 1.5$, D = 1.5 (YELLOW). Power thresholds are indicated by the red dashed line (80%) and the grey dashed line (90%). $R_0$, basic reproduction number, D, dispersion coefficient.

CI 71.9–88.1) power with $n_{rings}$ = 80. With $R_0$ = 1.5 and D = 1.5, the power was reduced substantially wherein $n_{rings}$ = 150 produced 62.8% power. The model for $R_0$ = 1.5 and D = 1.0 did not converge for any number of rings tested and is omitted from Fig 3. The results were generally consistent when $n_{sim}$ was varied.

## Sensitivity analyses

Applying a slower duration of implementation of 2 days meant that 80% power is reached with >125 rings. Using lower population coverage of 50% and 60% increased the sample size required to >100 rings to reach 80% power. Lowering slightly the population coverage to 75% resulted in 79.1% power reached with 100 rings, which is close to the target of 80% power.

## Discussion

Our simulation strategy provided a relatively simple means of estimating power and associated sample sizes for an observational study of CATI. Based on an $R_0$ of 2.0, the sample size required to reach 80% power was 80–100 rings, which was generally maintained when population coverage decreased from 80% to 75%. This would have been feasible during recent experiences in implementing CATI during large epidemics in Kathmandu Valley, Nepal (169 rings in 7 months), and Centre Department, Haiti (238 rings in 24 months). [27,28] Alternately, where CATI was used to suppress the tails of large outbreaks though only at the end of mass vaccination campaigns in Juba, South Sudan and Kribi, Cameroon, the sample size would far exceed the number of rings that are typically implemented. [19,20] As cholera epidemics frequently remain small due to the burn-out of the susceptible pool [18], overdispersion of $R_0$ leading to extinction [16,17], or the impact of the interventions themselves, a pooled analysis of multiple epidemics within a country implementing the same CATI package could be a more secure prospect to attain the required sample size.

A strength of this simulation strategy is the inclusion of a realistic depiction of CATI implementation which models the relative effects of its composite interventions over time. This accounted for the time-limited effects of antibiotics (~ 2 days) and the 7–11 day delay to a measurable immune response after administration of vaccination. [2,24] Another strength was that the stochastic model accounted for the depletion of susceptible persons to provide a plausible representation of early epidemic growth in a small population. This approach can be adjusted using the real-time estimates of the effective reproduction number ($R_E$) to update sample size estimation. It is also computationally-light, as the process takes less than 2 hours to run without the use of parallel computing.

There are key limitations to our methodology and its simplifying assumptions. The mean population size of 500 persons reflects urban settings. However, cholera epidemics can occur across urban and rural areas simultaneously and would include smaller rings with lower intracluster variation in incidence. As such, a larger sample size may be required to reach 80% power. For the stochastic model, several parameters relating to the early growth of a cholera epidemic are uncertain. The main model used a relatively high $R_0$ (2.0) sourced from early epidemic growth in unvaccinated populations in South Sudan and Yemen; considerably lower power was achieved with $R_0$ = 1.5. [14,15] In addition, we assumed the entire population was susceptible at the start of the outbreak, which may not be the case in cholera-endemic or previously-vaccinated areas, lowering the $R_E$ and the measurable effect of CATI. The stochastic model is not spatially-explicit, so transmission between communities is not accounted for, nor is the force of infection external to a given ring which could represent long-distance transmission from outside the community or contamination of the local water supply. [2] A duration of implementation of a single day has been shown in Cameroon and South Sudan [19,20], but

this may not be sufficient to cover the entire ring. This potentially leads to an overestimation of the effect, with the sensitivity analysis finding higher sample size requirements. Outbreak simulations are right censored at 30 days, and thus we cannot determine from the 30 day analysis alone whether outbreaks are fully extinct. How the delay to case detection was parameterized as a random effect may not truly represent the surveillance capacity, indicating that it must be accounted for empirically in the actual analysis of the study. Similarly, key co-variates that are thought to be influential on ring incidence (i.e., coverage, average rainfall, distance from roads) could not be simulated realistically without a more complex, spatially-explicit transmission model.

Despite its limitations, the strategy demonstrates a relatively simple and efficient approach to integrating dynamic modeling of a cholera outbreak with study simulation to guide the design of a prospective observational study that we intend to implement. The approach can be used to provide power estimates for evaluations of similar highly targeted interventions for epidemic-prone diseases delivered rapidly to high-risk communities during an outbreak.

## Supporting information

**S1 Text. Model fit results and sensitivity analyses.** Table A. Power estimates from main simulations and sensitivity analyses Fig A: power estimates using a duration of implementation of two days Fig B: power estimates using a population coverage of 50% Fig C: power estimates using a population coverage of 60% Fig D: power estimates using a population coverage of 75%.
(DOCX)

## Acknowledgments

We thank Sebastian Funk (LSHTM) for his advice on the bpmodels package.

## Author Contributions

**Conceptualization:** Ruwan Ratnayake, Francesco Checchi, W. John Edmunds, Flavio Finger.

**Data curation:** Ruwan Ratnayake.

**Formal analysis:** Ruwan Ratnayake, Flavio Finger.

**Funding acquisition:** Ruwan Ratnayake.

**Investigation:** Ruwan Ratnayake, Flavio Finger.

**Methodology:** Ruwan Ratnayake, Francesco Checchi, Christopher I. Jarvis, W. John Edmunds, Flavio Finger.

**Project administration:** Ruwan Ratnayake.

**Resources:** Ruwan Ratnayake.

**Software:** Ruwan Ratnayake, Flavio Finger.

**Supervision:** Francesco Checchi, W. John Edmunds, Flavio Finger.

**Validation:** Ruwan Ratnayake, Flavio Finger.

**Visualization:** Ruwan Ratnayake.

**Writing – original draft:** Ruwan Ratnayake, Flavio Finger.

**Writing – review & editing:** Francesco Checchi, Christopher I. Jarvis, W. John Edmunds, Flavio Finger.

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
