## [Decision Letter · Decision Letter 0]

17 Nov 2021

Dear Dr. Rowan Ratnayake,

Thank you very much for submitting your manuscript "Inference is bliss: Simulation for power estimation of a cholera outbreak intervention study (short report)" for consideration at PLOS Neglected Tropical Diseases. As with all papers reviewed by the journal, your manuscript was reviewed by members of the editorial board and by several independent reviewers. In light of the reviews (below this email), we would like to invite the resubmission of a significantly-revised version that takes into account the reviewers' comments. 

Sorry for delayed decision due to the difficulty to recruit the reviewers. As you can see the two reviewers gave the different opinions about your MS. Please revise the MS according the reviewers' comments.

We cannot make any decision about publication until we have seen the revised manuscript and your response to the reviewers' comments. Your revised manuscript is also likely to be sent to reviewers for further evaluation.

Sincerely,

Ruifu Yang

Deputy Editor

Ruifu Yang

Deputy Editor

Sorry for delayed decision due to the difficulty to recruit the reviewers. As you can see the two reviewers gave the different opinions about your MS. Please revise the MS according the reviewers' comments.

Reviewer's Responses to Questions

**Key Review Criteria Required for Acceptance?**

**Methods**

-Are the objectives of the study clearly articulated with a clear testable hypothesis stated?

-Is the study design appropriate to address the stated objectives?

-Is the population clearly described and appropriate for the hypothesis being tested?

-Is the sample size sufficient to ensure adequate power to address the hypothesis being tested?

-Were correct statistical analysis used to support conclusions?

-Are there concerns about ethical or regulatory requirements being met?

Reviewer #1: (No Response)

Reviewer #2: Yes

**Results**

-Does the analysis presented match the analysis plan?

-Are the results clearly and completely presented?

-Are the figures (Tables, Images) of sufficient quality for clarity?

Reviewer #1: (No Response)

Reviewer #2: Yes

**Conclusions**

-Are the conclusions supported by the data presented?

-Are the limitations of analysis clearly described?

-Do the authors discuss how these data can be helpful to advance our understanding of the topic under study?

-Is public health relevance addressed?

Reviewer #1: (No Response)

Reviewer #2: Yes

**Editorial and Data Presentation Modifications?**

Reviewer #1: (No Response)

Reviewer #2: (No Response)

**Summary and General Comments**

Reviewer #1: This manuscript describes a general framework including mathematical modeling and simulation, to evaluate outbreak containment for pathogens in geographically or socially defined rings. It focuses on the size of samples require to achieve 80%-90% power in regression analysis, to an evaluation of ring vaccination. A data-generating mechanism is a stochastic transmission model to predict the incidence and several parameters are added to the stochastic transmission model to simulate realities. The main finding of the manuscript is that targeting 80 to 100 rings was required to achieve power ≥80%, with a basic reproduction number of 2.0 and a dispersion coefficient of 1.0—1.5.

The authors make detailed assumptions in the stochastic transmission model. While I do have two concerns regarding the reported finding. 

The first one I deem highly important, about the simulation method. Only a generalized linear mixed model (GLMM) was used to estimate the response variable, it always makes good simulations and inferences. But I was also curious whether the finding showed similar results under different simulation models. Due to different scenarios and the habits of researchers, the conclusions obtained in the GLMM model are not necessarily applicable to guide other models; For example, under more powerful simulation techniques such as popular machine learning models, whether the need for sample sizes will decrease?

Secondly, in the section of the stochastic transmission model (Line 209), the authors set an assumption about CATI implementation cost and population-based coverage. What are the reasons for not using a normal distribution simulation in the experiment? Similar to the conclusion of delay to CATI in Fig.2, what impact will it have on the results to reduce the required time of CATI and cover more people? It was necessary to try to simulate different situations about the population-based coverage to consider different action.

Finally, as the author mentioned, some key variables are not included by the stochastic model, and I realized it is an obstacle that is hard be overcome, it does lower the strength of the analysis. Nevertheless, I do think that finding is a relatively conservative and valid conclusion.

Reviewer #2: The study by Ratnayake et al. aimed at performing a simulation for power estimation of a cholera outbreak intervention, describing a strategy combining mathematical modelling and simulation for that purpose. They found that the implementation of the estimation strategy evaluated is practicable for observational study designs that aim to evaluate the containment of outbreaks occurring within geographically delineated clusters. The manuscript is well written and it has been performed with an adequate methodology. The results support the conclusions.

PLOS authors have the option to publish the peer review history of their article (what does this mean?). If published, this will include your full peer review and any attached files.

Reviewer #1: No

Reviewer #2: No
---

## [Decision Letter · Decision Letter 1]

11 Jan 2022

Dear Dr. Ruwan Ratnayake,

We are pleased to inform you that your manuscript 'Inference is bliss: Simulation for power estimation for an observational study of a cholera outbreak intervention (short report)' has been provisionally accepted for publication in PLOS Neglected Tropical Diseases.

Best regards,

Ruifu Yang

Deputy Editor

Ruifu Yang

Deputy Editor

Reviewer's Responses to Questions

**Key Review Criteria Required for Acceptance?**

**Methods**

-Are the objectives of the study clearly articulated with a clear testable hypothesis stated?

-Is the study design appropriate to address the stated objectives?

-Is the population clearly described and appropriate for the hypothesis being tested?

-Is the sample size sufficient to ensure adequate power to address the hypothesis being tested?

-Were correct statistical analysis used to support conclusions?

-Are there concerns about ethical or regulatory requirements being met?

Reviewer #1: (No Response)

Reviewer #3: (No Response)

**Results**

-Does the analysis presented match the analysis plan?

-Are the results clearly and completely presented?

-Are the figures (Tables, Images) of sufficient quality for clarity?

Reviewer #1: (No Response)

Reviewer #3: (No Response)

**Conclusions**

-Are the conclusions supported by the data presented?

-Are the limitations of analysis clearly described?

-Do the authors discuss how these data can be helpful to advance our understanding of the topic under study?

-Is public health relevance addressed?

Reviewer #1: (No Response)

Reviewer #3: (No Response)

**Editorial and Data Presentation Modifications?**

Reviewer #1: (No Response)

Reviewer #3: (No Response)

**Summary and General Comments**

Reviewer #1: The authors have well addressed my concerns.

Reviewer #3: After reading the manuscript, I feel I am not an expert in this field to comment or review. So, I am sorry to review it.

Thanks

PLOS authors have the option to publish the peer review history of their article (what does this mean?). If published, this will include your full peer review and any attached files.

Reviewer #1: No

Reviewer #3: No

---

## [Editor Report · Acceptance letter]

11 Feb 2022

Dear Mr. Ratnayake,

We are delighted to inform you that your manuscript, "Inference is bliss: Simulation for power estimation for an observational study of a cholera outbreak intervention," has been formally accepted for publication in PLOS Neglected Tropical Diseases.

Best regards,

Shaden Kamhawi

co-Editor-in-Chief

Paul Brindley

co-Editor-in-Chief
